# Effect of Light Intensity and Light Spectrum of LED Light Sources on Photosynthesis and Secondary Metabolite Synthesis in *Ocimum basilicum*

**DOI:** 10.3390/plants14091334

**Published:** 2025-04-28

**Authors:** Luca Jokic, Isabell Pappert, Tran Quoc Khanh, Ralf Kaldenhoff

**Affiliations:** 1Department of Applied Plant Sciences, Faculty of Biology, Technical University Darmstadt, 64287 Darmstadt, Germany; isabell.pappert@tu-darmstadt.de (I.P.); kaldenhoff@bio.tu-darmstadt.de (R.K.); 2Department of Adaptive Lighting Systems and Visual Processing, Technical University Darmstadt, 64287 Darmstadt, Germany; khanh@lichttechnik.tu-darmstadt.de

**Keywords:** basil, photosynthesis, gas exchange, chlorophyll fluorescence, light conditions, total phenolic content, total flavonoid content, energy fluxes

## Abstract

Basil is best known as an aromatic and medicinal herb due to its rich profile of bioactive compounds. While secondary metabolite production, coupled with growth, has been well studied, photosynthesis has often been overlooked in this regard. In this study, we investigate the effect of light intensities of blue, green, red, and white light of semiconductor LEDs up to 10000 µmol m^−2^ s^−1^ on photosynthetic efficiency and primary and secondary metabolism. Chlorophyll fluorescence data indicate that the conversion of light into chemical energy is the same under green, red, and white light, and 35% increased under blue light. Primary metabolism, represented by assimilation rate, shows that blue light has the lowest assimilation, whereas red and (surprisingly) green light have the highest. Light saturation is reached at 1500 µmol m^−2^ s^−1^, while assimilation under green light is maintained up to 5000 µmol m^−2^ s^−1^. The earliest photoinhibition occurred under blue light in comparison to the other light wavelength under investigation. Blue light also enhances the production of phenolic and flavonoid concentrations up to 40% or 100%, respectively. These results show that photosynthesis, photoinhibition, and secondary metabolite production are wavelength-dependent and indicate how energy fluxes between these processes are related.

## 1. Introduction

*Ocimum basilicum* L. (basil), also known as the “King of Herbs”, comes from India and South Asia and is native to tropical to subtropical regions [1,2]. In recent decades, it has gained popularity not only in food and cosmetic industries but also for its health benefits [2,3,4].

The versatility of basil is due to its rich profile of bioactive compounds, such as oils, phenolics, and flavonoids. Basil extracts have antioxidant properties, helping to reduce cell damage by free radicals [5]. These extracts are anti-microbial [6,7,8,9,10,11], anti-inflammatory [12,13,14,15], anti-diabetic [16,17], or anti-cancerogenic [18,19,20,21,22]. Basil has also been linked to positive effects on memory [23].

In addition to its positive and marketable aspects, growing basil is rewarding due to rapid growth. Depending on cultivar and environmental conditions, the vegetation period can range from 50 to 90 days [24]. It is also common practice to harvest some of the basil prior to flowering. This multi-cut harvesting strategy reduces harvesting time to as little as 40 days after sowing [25] and is believed to significantly increase yields [3,26]. Taken together, these factors contribute to basil being classified as a “high value crop” [27]. Independent reports predict annual growth of the basil market by 3.4 to 4.1% until 2028/2033 [28,29].

Thus, basil growth and its production of secondary metabolites in response to light has been studied extensively. Optimal light conditions depend on two key factors: light intensity and duration, combined as daily light integral (DLI), and spectral composition. It has been shown that basil growth and secondary metabolite production correlate with increasing DLI up to an optimum [30,31,32,33,34]. A further increase in biomass or secondary metabolites can be achieved by changing the spectral composition from white light to a combination of blue and red light [35,36,37,38]. However, the optimal ratio of blue to red light has also been the subject of much debate. While some studies claim that a higher proportion of blue light promotes growth and secondary metabolites [39], there are conflicting studies claiming the opposite [40]. Another study showed that while one spectral combination may be beneficial for one cultivar, the opposite combination may be beneficial for another [41]. The optimal spectral combinations identified are often limited by the specific environmental factors of the study and, in particular, the cultivars used. The comprehensive review by Sipos [27] highlights the wide range of research focusing on the effects of light intensity, duration, and spectral composition on basil growth and secondary metabolite production.

In contrast, the influence of light intensity and light spectrum on photosynthesis has not been studied comprehensively with respect to basil. Photosynthesis/gas exchange measurements were mainly used to evaluate results from growth experiments. It was demonstrated that the assimilation rate, transpiration rate, stomatal conductance, and stomatal CO_2_ concentration increase up to a certain DLI [30,31]. In addition, light spectrum influences both plant growth and photosynthetic potential. Green basil grown under a 3:1 ratio of red and blue light showed the highest assimilation rate compared to white light [42]. In two more cultivars, the same ratio of 3:1 resulted in the highest assimilation rate, while the transpiration rate was increased with an inversed ratio of 1:3 (high proportion of blue light) [41]. Further studies, moreover, showed that basil grown under red–blue light ratios of 1:2 or 1:3 led to an increase in maximum assimilation rate and stomatal conductance in two different cultivars [35]. Taken together, key factors that affect photosynthetic performance include cultivar [43], leaf level [42], and plant age, but also growing conditions such as lighting, nutrient and water management [44,45,46], or growing temperature [47,48].

Despite these multivariate influences, photosynthesis in basil follows a similar pattern, as determined from the results of different scientific studies. Coutinho [43] showed that basil cultivars may have different maximum assimilation rates (27.75 for the green cultivar and 23.96 for the red cultivar; *Ocimum basilicum* L.), but they follow the same photosynthesis saturating curves, with a maximum near 1500 µmol m^−2^ s^−1^. With increasing light intensity, a plateau can be observed that continues up to the maximum light intensity tested. Findings from Park [49] support these results. Here, a saturating light response curve with a maximum at 1500 µmol m^−2^ s^−1^ was observed as well. It was found to be limited to light intensities higher than that of sun-light (2000/2500 µmol m^−2^ s^−1^). Both studies were conducted with a warm white spectrum composed of a mixture of blue, green, and a high proportion of red light. The effect of monochromatic light on photosynthesis was not investigated. It has long been known that photosynthesis is wavelength-dependent. Photosynthetic pigments effectively absorb blue and red light, with maximum absorption peaks at 430–450 and 640–660 nm, respectively [50]. This is the reason for the misconception that blue and red light drive photosynthesis most efficiently. In the case of the top few cell layers of a leaf, this conception could be applicable because the bulk of red and blue light is generally absorbed here. When we consider the entire leaf, the circumstances change. Because of sieve and detour effects, the absorption of green light is much greater than generally assumed [51]. McCree postulated that at low PPFDs, the quantum yield of green light is higher than that of blue light and only slightly lower than that of red light [52].

Using a recently developed set of high intensity LED modules, we were able to apply light intensities of almost 10,000 µmol m^−2^ s^−1^ with four different light colors (red, blue, green, and warm white light). The maximum light intensity produced by these LED-systems is about 4–5 times that of sunlight intensity for each light color and about 100 times higher compared to the studies mentioned above. The question arises as to whether plants are able to utilize the high-intensity light for photosynthesis, or if increased absorption, combined with excess energy, promotes photodamage. Given that the energy of photons is inversely proportional to wavelength [53], disadvantageous effects were expected to be observed for blue light first.

Unexpectedly, our studies revealed that basil can indeed use high light intensities for photosynthesis. The results described here expand our view on photosynthesis and open new possibilities for cultivation.

## 2. Results

To understand the relationship between energy conversion and allocation to primary and secondary metabolism depending on wavelength and light intensity, three assessment methods were applied under the same experimental light conditions. First, the conversion efficiency of light energy into chemical energy was determined using chlorophyll analysis. These data were used to determine the utilization of chemical energy in primary metabolism by measuring the assimilation rate via a gas exchange system and the production of secondary metabolites via the photometric determination of phenols and flavonoids.

### 2.1. Chlorophyll Fluorescence: Maximum Electron-Transport-Chain Efficiency with Blue Light

The quantum yield curve for white, red, green, and blue light (Figure 1) follows a negative exponential curve and reaches a plateau value of 0.13 at 7000 µmol m^−^^2^ s^−^^1^. In the range from 0 to 2000 µmol m^−^^2^ s^−^^1^, quantum yield under white, red, and green light is similar, as is that of white and green from 0 to 7900 µmol m^−^^2^ s^−^^1^. As red light overrides chlorophyll fluorescence, quantum yield under red light could only be monitored up to 2500 µmol m^−^^2^ s^−^^1^. Quantum yield under blue light differs significantly between 800 and 5600 µmol m^−^^2^ s^−^^1^. From 2500 to 3500 µmol m^−^^2^ s^−^^1^, blue light is converted into chemical energy 36% more efficiently compared to the other light spectral conditions tested.

The curves for non-photochemical quenching (Figure 2) are mirror-inverted compared to those of quantum yield. Instead of a negative exponential graph, a positive saturation curve can be observed. All four illuminants start from the same point and approach a plateau value between 0.65 and 0.75 at the end of the saturation curve. There are no significant differences between the white, red, and green light in a range from 0 to 2500 µmol m^−^^2^ s^−^^1^ and between white and green up to 5000 µmol m^−^^2^ s^−^^1^. Comparable to quantum yield data, non-photochemical quenching of blue light significantly differs, emitting the least amount of heat in a range from 800 to 5600 µmol m^−^^2^ s^−^^1^.

### 2.2. Gas Exchange: Green and Red Light Outperform Blue Light Despite Its Energy Abundance

The assimilation rates of the four illuminations start in the dark at −0.26 µmol m^−^^2^ s^−^^1^ and show a steep increase up to their light saturation points (Figure 3). The highest maximal assimilation can be observed for green and red light with 11.2 µmol m^−^^2^ s^−^^1^ at 4000 µmol m^−^^2^ s^−^^1^ and 2500 µmol m^−^^2^ s^−^^1^, respectively. White reaches its maximum at 2900 µmol m^−^^2^ s^−^^1^ with 9.1 µmol m^−^^2^ s^−^^1^. Blue light shows the lowest assimilation with 8.9 µmol m^−^^2^ s^−^^1^ at 1500 µmol m^−^^2^ s^−^^1^. Unexpectedly, white and red light lead to a plateau in assimilation starting at 1500 µmol m^−^^2^ s^−^^1^, and for green light, the plateau starts at 2900 µmol m^−^^2^ s^−^^1^. For white light, this plateau lasts until 4000 µmol m^−^^2^ s^−^^1^; for green light, it last until 5000 µmol m^−^^2^ s^−^^1^; and for red, it lasts light until 3500 µmol m^−^^2^ s^−^^1^. As soon as these intensities are exceeded, photoinhibition and a distinct decrease in assimilation rates can be observed, with blue light being the first light condition to experience this. 

A temperature of 24.4 degrees was the initial leaf temperature (Figure 4). Increasing light intensity beyond 550 µmol m^−^^2^ s^−^^1^ led to a rapid temperature increase for all illumination conditions. At maximum illumination, the temperature rose to 27.2 °C for green light, 28.4 to 29.4 °C for red and white light, and 31 °C for blue light. Green light causes the smallest increase in temperature, and blue light causes the largest, indicating that it promotes the most excess energy that cannot be used for photosynthesis and is therefore emitted as heat. Additionally, blue light enables the highest stomatal opening, which consequently also allows for the highest water evaporation (Appendix A).

### 2.3. Secondary Metabolites: How Intensity and Exposure Time of Blue Light Enhances Its Production

A series of experiments were conducted to explore how different light spectra (light colors) affect total phenolic and flavonoid content. From preliminary experiments, it was hypothesized that a 60 min exposure time at 500 µmol m^−^^2^ s^−^^1^ was sufficient to significantly increase both phenolic and flavonoid concentrations. Since the initial adaptation phase of the gas exchange measurements also took place at an intensity of 600 µmol m^−^^2^ s^−^^1^ and a duration of 45 min, it was first investigated as to whether even a very short exposure of <60 min at a light intensity of 500 µmol m^−^^2^ s^−^^1^ of either light condition could lead to differences in phenolic and flavonoid concentrations in the directly illuminated, upper leaves, between plants grown under greenhouse conditions and plants illuminated with white, green, or red light for 45 min at 500 µmol m^−^^2^ s^−^^1^ (Figure 5A). While these illuminations resulted in a phenol concentration of 1.81 to 1.96 mg gallic equivalents per g of frozen leaf powder (GAE), blue light led to a total concentration of 2.31 mg/g GAE, which significantly differs from greenhouse-grown plants and plants irradiated with white and green light. Total flavonoid content, expressed as mg catechin equivalents per g of frozen leaf powder (CE), also revealed no significant differences between the illumination of white, green, or red light and greenhouse-grown plants (Figure 5B). Irradiation with blue light is again an exception; in this instance, the concentration of flavonoids is significantly increased to a value of 1.63 mg/g CE, which is a difference of 85% in comparison to greenhouse-grown plants.

A follow-up experiment was conducted to determine phenolic and flavonoid concentrations at the point where the greatest differences in assimilation rate and quantum yield occur between illuminations to see if this increase could be further enhanced. This would be in a range of 2000–2500 µmol m^−^^2^ s^−^^1^. Illumination with increasing light intensities up to 2500 µmol m^−^^2^ s^−^^1^ for 90 min intensifies the previously observed phenomenon of 45 min of 500 µmol m^−^^2^ s^−^^1^ illumination. While no significant differences in the concentration of phenolics or flavonoids were observed between green, red, white light, and greenhouse-grown plants, blue light increased the concentration of phenolics from 25 to 40% and the concentration of flavonoids from 85 to 100% (N = 10 for each light color) (Appendix A).

### 2.4. Blue Light Does Not Induce a Uniform Increase in Phenol- or Flavonoid Concentration Throughout the Plant

For a third experiment, it was investigated whether the altered concentration of phenolics and flavonoids in the upper illuminated leaves is only a local effect or a systemic response of the whole plant.

A significant difference in phenol concentration can be seen when comparing the upper to the middle or lower leaf level of the greenhouse-grown plants (Figure 6A). In these plants, phenol concentration in the leaf layers is down-regulated as distance from the light source increases. The middle and lower leaf layers of greenhouse-grown plants showed no significant differences from each other or from the other middle leaf layers of the four different illuminations, although a slight increase in phenolic concentration can be seen after blue light illumination. Their concentration ranged from 1.20 to 1.30, with blue light being the exception at 1.47 mg/g GAE. Phenol concentration varies across leaf layers, with lower layers containing a significantly lower concentration. There were no significant differences observed in the flavonoid concentration between the different leaf levels of the greenhouse-grown plants, nor between the middle, less-illuminated leaf levels of the previous irradiation experiment (Figure 6B).

## 3. Discussion

By comparing data obtained via chlorophyll fluorescence analysis, gas exchange measurements, or photometric determination of secondary metabolites, this study provides an integrative approach to comprehend energy conversion and fluxes in primary and secondary metabolism under different light wavelengths and light intensities in basil. At the same time, the limits of the photosynthetic apparatus at light intensities of 4 to 5 times that of solar radiation are demonstrated.

### 3.1. Light Saturation Curves Under Different Light Colors

CO_2_ assimilation increases with light intensity up to a point of 1500 µmol m^−2^ s^−1^. Beyond this light saturating point, further increases in intensity did not lead to a significant increase in assimilation, except for green light. Instead, a plateau is observed before assimilation decreases due to photo-inhibitory effects. This has also been seen in previous works where maximal assimilation was observed at 1500 µmol m^−2^ s^−1^. Here, we went beyond the limits of these studies, which describe a plateau up to 2500 µmol m^−2^ s^−1^ under white light [43,49], and demonstrate that the plateau continues up to a light intensity of 4000 µmol m^−2^ s^−1^ prior to a significant drop in assimilation. A further difference lies in the absolute maximum assimilation rates observed, which are more than twice as high as in the present work. Several reasons may explain this difference. First, different basil cultivars were used. Second, the spectral distribution of the measurement light was not mentioned; white light has a range from 1500 to 8000 Kelvin, and each wavelength has a different effect on assimilation, as clearly demonstrated in this work. But one major difference was in the cultivation conditions. Park [49] uses a hydroponic system for cultivation, with a light spectrum of 3:7 blue to red at an intensity of 150 µmol m^−2^ s^−1^, while Cuthino [43] grew his plants outdoors without an adjustable source of light. The significant influence of cultivation light on assimilation has already been shown, with higher intensities/DLIs correlating to higher assimilation rates up to a certain point [30,31].

### 3.2. Green Light Drives Photosynthesis More Effectively Than Other Light-Colors

Unexpectedly, green light does not reach saturation until 2900 µmol m^−2^ s^−1^, maintaining its plateau at light intensities up to 5000 µmol m^−2^ s^−1^. We detected maximum assimilation comparable to red light even though it is less absorbed and partially reflected, dissimilar to the bulk of red or blue light [54]. It is known that 10 to 50% of green light is reflected [51]. Plants grown with higher nitrogen levels, which develop a denser and more robust photosynthetic apparatus, characterized by more thylakoids and chlorophyll, enhance the absorbance of green light, while the absorption of blue and red light remains unaltered [55]. Chlorophyll a and b are responsible for the absorption of most of the light, and the absorption spectrum of a plant is largely defined by these pigments. In leaf tissue, chlorophylls are often associated with other pigments such as xanthophylls and carotenoids. These pigments are photoprotective and contribute to non-photochemical quenching. Nevertheless, some of the energy absorbed by these pigments is transferred to chlorophyll and subsequently used in photochemical reactions. In this way, carotenoids extend the absorption spectrum into the green region [56,57,58,59].

While these studies emphasize that the absorption of green light is significant, its effectiveness turned out to be lower than that of blue and red light. However, while 90% of blue and red light is usually absorbed in the upper 20% of the leaf [60], green light can penetrate deeper into leaf tissue. This phenomenon is also reflected in CO_2_ fixation, whereas most of CO_2_ fixation driven by green light occurs in the middle-to-lower cell-layers of the leaf [61]. A major reason for this is the so-called detour effect, which causes increased scattering of green light within the leaf. As a result, light is more evenly distributed, and photosynthesis is more homogeneous, particularly in deeper leaf layers [51]. This combined effect—deeper penetration and better distribution—leads to the green light effect, specifically at high light intensities. In lettuce, it has been demonstrated that red light results in higher assimilation rates at low light intensities due to its higher absorption, whereas green light leads to higher assimilation rates beyond a certain light intensity [62]. Another advantage due to the low absorption of green light is its thermal effect: compared to blue and red light, green light causes less heating of the leaf. This minimizes the thermal overload and potential oxidative damage that occurs with intensive illumination of red or blue light. Accordingly, green light triggers fewer photo inhibitory processes due to its lower absorption.

### 3.3. Blue Light Triggers Photo Inhibitory Processes First

The lowest maximal assimilation and the earliest decrease in assimilation is seen under blue light. This decrease is observable under all illuminations and can be explained by typical photo inhibitory phenomena. These include the formation of reactive oxygen species, the photooxidation of pigments, or even damage to the chloroplast structure [63,64,65,66]. As blue light triggers photo inhibitory processes, its effect is also modified. Studies support the concept of wavelength-specific photoinhibition. In plants such as beans or pumpkins, it has been shown that UV light and blue light cause stronger photoinhibition than the other wavelengths of visible light [67]. Similar results can be found in micro-algae. The highest rate of photoinhibition was found in the range from UV to blue light [68].

### 3.4. Blue Light Triggers Secondary Metabolism

Chlorophyll fluorescence analysis in 2.1 shows that there are no differences in the energy conversion of green, red, or white light into chemical energy. Only blue light is more effectively processed. Despite blue light having the most efficient energy conversion, only a portion of its energy is utilized by the plant for assimilation. Since blue light triggers photo-inhibitory processes, a hypothesis was formulated and tested suggesting that the surplus energy is directly used to prevent damage to the photosynthetic apparatus through the formation of secondary metabolites. This hypothesis is based on the observation that phytopathogens reduce primary metabolism, especially photosynthesis and chlorophyll production, to provide more energy for the plant’s energy consuming defense response [69]. UV light and blue light are often considered to be stress factors because of the high energy of the photons, which promotes the formation of free radicals [70]. To prevent possible damage from free radicals, antioxidants such as phenols and flavonoids are formed, which are among the most important secondary metabolites [71,72]. This formation, as with other defensive responses of plants, is energy consuming, and as a result, primary metabolism could be reduced, which results in a reduced assimilation rate in comparison to other light conditions. Our analysis of phenolic and flavonoid concentrations supports this hypothesis. Following the blue light irradiation, a significant increase in secondary metabolites was observed after short, low light intensity, as well as with longer, successively increasing light intensity. However, this effect seemed to be spatially limited.

## 4. Conclusions

We showed that blue light irradiation with a peak wavelength of 450 nm and with a photon density of 500 µmol m^−2^ s^−1^ for only 45 min is sufficient to increase the phenolic concentration by 25% and the flavonoid concentration by 85%. These concentrations can be further increased through longer exposure times and higher intensities. By applying this short-time, low-energy treatment to plants as an end-of-production/pre-harvest treatment, their antioxidant content could be significantly and lastingly increased such that the food industry, the pharmaceutical industry, or even the consumer could benefit directly. As a post-harvest treatment, an increase in antioxidant levels could improve shelf life by delaying senescence and preserving both the nutritional value and the overall quality of the fresh product.

In addition, this study showed that both photosynthesis and photoinhibition are wavelength- and light intensity-dependent and how the energy distribution within a plant between primary and secondary metabolism might look like. The most efficient energy conversion of blue light does not induce the highest assimilation rate, and although green, red, and white light have the same efficiency in converting light energy into chemical energy, they have different assimilation rates, suggesting that the energy ends up in other processes within the plant.

## 5. Materials and Methods

### 5.1. Plant Cultivation

*Ocimum basilicum* L., with the cultivar Genoversa purchased from Kniepenkerl a trademark of Bruno Nebelung GmbH, 48351 Everswinkel, Germany, was cultivated under greenhouse conditions with a day/night cycle of 12/12 h at an average temperature of 21.5 °C and a humidity of 80%. LED lights with a ratio of white to red light of 3:1 and a light intensity of 200 to 300 µmol m^−^^2^ s^−^^1^, depending on the plant height, were used as the light source. Plants were used only once for each of the following three methods at an age between 20 and 45 days after sowing. One of the youngest but fully developed upper leaf pairs was used for chlorophyll fluorescence and gas exchange analysis. For photometric analysis of secondary metabolites, upper, middle, and lower leaves were used (Figure 7).

### 5.2. Chlorophyll Fluorescence Analysis

A MAXI version of the IMAGING-PAM M series was used to measure chlorophyll fluorescence. More precisely, an IMAG-MAX/L with an IMAG-K7 camera from Heinz Walz GmbH, Effeltrich, Germany, was used. Prior to measurement, the plants were incubated in the dark overnight. After this dark incubation period, a light curve was recorded. This light curve started with a 15 min adaptation period to the lowest light intensity of the respective wavelength, after which the light intensity was increased every eight minutes. Each light intensity level was only increased once the plant/chlorophyll fluorescence had reached a steady state. For each level of light intensity, the mean value of the last 1.5 min was used for evaluation. This procedure was repeated ten times for each wavelength. The quantum yield of photosystem II and non-photochemical quenching were calculated from chlorophyll fluorescence. Both the dark incubation and the measurements were carried out in a dark phytochamber under the same greenhouse conditions, except for light exposure. The external light source used to illuminate the leaf was an LED module from Chips 4 Light GmbH, Sinzing, Germany.

### 5.3. Gas Exchange Analysis

A GFS-3000 gas exchange system from Heinz Walz GmbH was used to generate additional parameters such as assimilation rate, transpiration rate, stomatal conductance, and leaf temperature. As with the previous measurement method, a light curve was also recorded in a dark phytochamber. The measuring procedure and the evaluation follow the same principle as with the IMAGING-PAM. After dark incubation, adaptation to the external light source lasted 45 min. The light intensity was then increased every 20 min after reaching a steady state level, and the mean value of the last two minutes was used for evaluation. The measurement inside the cuvette was carried out at a CO_2_ concentration of 400 ppm, a humidity of 18,000 ppm, and a temperature of 25 °C.

### 5.4. Photometric Analysis of Secondary Metabolites

In a follow-up experiment, plants were illuminated in a dark phytochamber following the same light program as described in Section 5.3. The plants analyzed were divided into three leaf levels: the upper leaves (youngest, directly illuminated leaves), the middle leaves, and the lower leaves, which were the oldest. Total phenolic and total flavonoid content were determined photometrically using a FoodALYT photometer from Omnilab-Laborzentrum GmbH and Co. KG, Bremen, Germany, following the protocols of Waterhouse [73] and Dou [30], with modified volumes. All chemicals used in this analysis were purchased from Carl Roth GmbH and Co. KG, Karlsruhe, Germany. 

#### 5.4.1. Extraction

Leaves were harvested after illumination and immediately frozen in liquid nitrogen. The leaves were then grinded to a fine powder in a mortar and pestle while still frozen. A total of 200–300 mg of this leaf powder was transferred to 1 mL of 80% methanol and incubated overnight at room temperature on an orbital shaker. After incubation, the mixture was centrifuged at 12,500 rpm for 30 min, and the supernatant was collected for further analysis.

#### 5.4.2. Total Phenolic Content (TPC)

Total phenolic content was determined according to the Waterhouse protocol with modified volumes. A 50 µL sample was mixed with 225 µL water and 125 µL Folin–Ciocalteu reagent, followed by incubation in the dark for 7 min. Then, 625 µL of 10% Na_2_CO_3_ solution was added, and the samples were incubated in the dark for a further 20 min. Absorbance was finally measured at 735 nm using a photometer. The results were expressed as gallic acid equivalents (GAE) per gram of frozen leaf powder (FLP).

#### 5.4.3. Total Flavonoid Content (TFC)

Total flavonoid content was determined following the protocol of Dou with modified volumes. A 100 µL sample was mixed with 500 µL water and 30 µL NaNO_2_. After 5 min incubation in the dark, 60 µL AlCl_3_ was added, followed by another 5 min incubation in the dark. Then, 200 µL of NaOH and 110 µL of water were added. Absorbance was measured at 510 nm, and the results were expressed as catechin equivalents (CE) per gram of frozen leaf powder (FLP).

### 5.5. Light Source

Seven-engine modules from Chips 4 Light GmbH were used as the external light source in all experiments. These high-power light modules are designed with a precise beam angle of only 10 degrees, ensuring focused and uniform illumination. With a wide spectral range from 367 nm to 940 nm and an advanced thermal management system capable of delivering light intensities of up to 10,000 µmol m^−^^2^ s^−^^1^, they are an essential and unique light source for investigating all types of plant responses to light. The spectrum of these modules, measured with a SpectraPen mini from PSI (Photon Systems Instruments) spol. s r.o., Drásov, Czech Republic, can be seen in the following illustration (Figure 8). The light intensities were assessed with a ULM-500 Light Meter and Logger with a Cosine Corrected Mini Quantum Sensor MQS-B and a Spherical Micro Quantum Sensor US-SQS/L from Heinz Walz GmbH.

## Figures and Tables

**Figure 1 plants-14-01334-f001:**
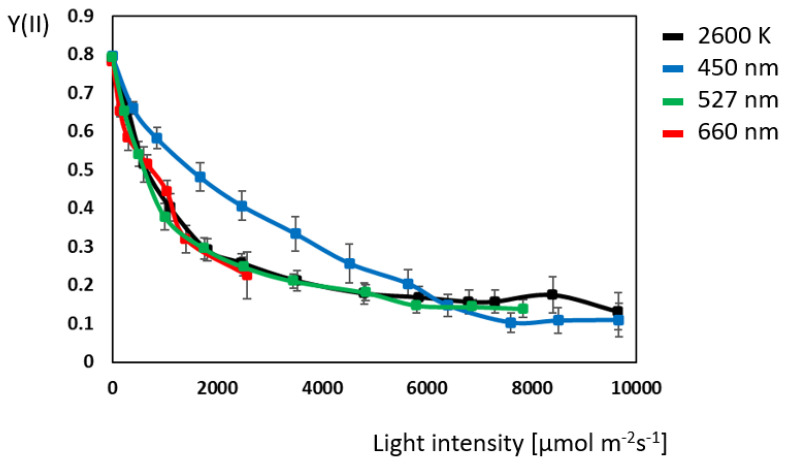
Comparison of photosystem II quantum yield (Y(II)) at different illuminations in *Ocimum basilicum* L. (N = 10 for each light color).

**Figure 2 plants-14-01334-f002:**
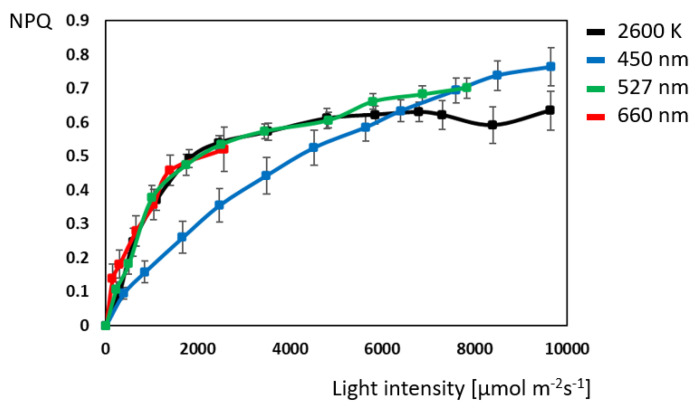
Comparison of non-photochemical quenching (NPQ) at different illuminations in *Ocimum basilicum* L. (N = 10 for each light color).

**Figure 3 plants-14-01334-f003:**
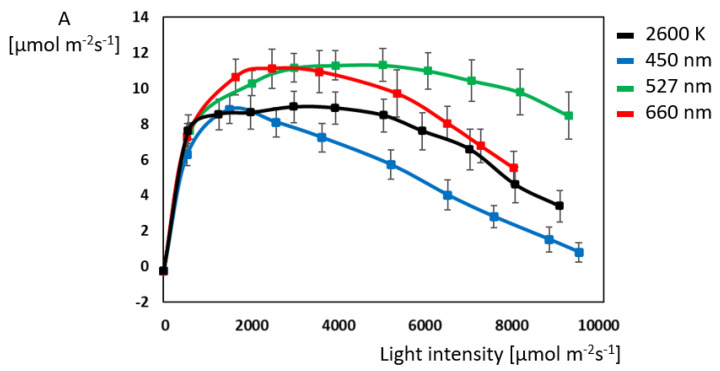
Comparison of assimilation rate (A) at different illuminations in *Ocimum basilicum* L. (N = 10 for each light color).

**Figure 4 plants-14-01334-f004:**
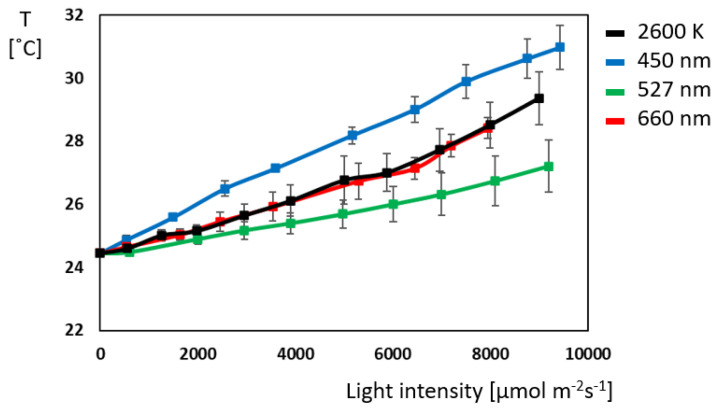
Comparison of leave temperature (T) at different illuminations in *Ocimum basilicum* L. (N = 10 for each light color).

**Figure 5 plants-14-01334-f005:**
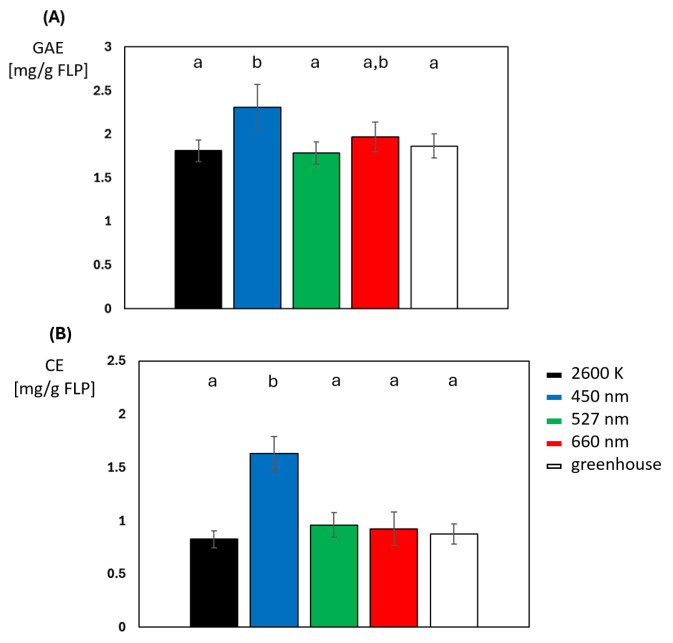
Comparison of (**A**) total phenolic and (**B**) total flavonoid content expressed in mg gallic equivalents and mg catechin equivalents per g of frozen leaf powder of greenhouse-grown plants and plants illuminated for 45 min at 500 µmol m^−2^ s^−1^ (N = 10 for each light color). Lower case letters indicate *t*-test calculated statistical significance when *p* < 0.05 was assumed.

**Figure 6 plants-14-01334-f006:**
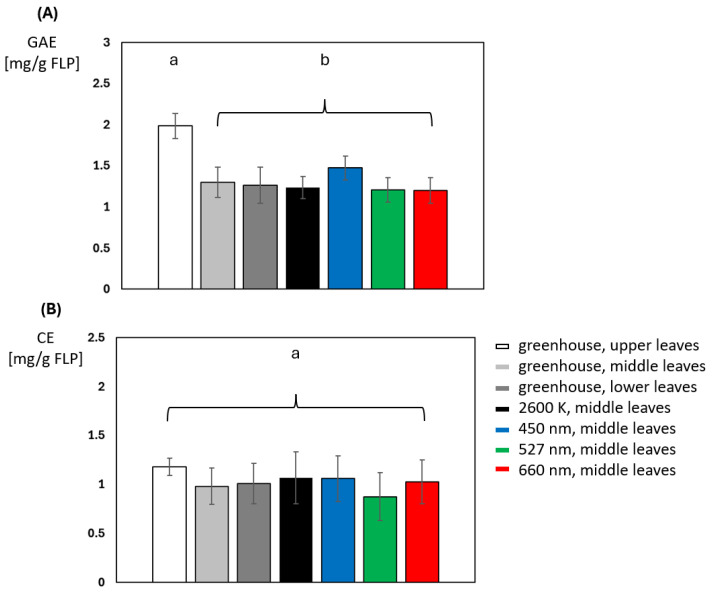
Comparison of (**A**) total phenolic and (**B**) total flavonoid content expressed in mg of gallic equivalents and mg catechin equivalents per g of frozen leaf powder of greenhouse-grown plants with leaves of different levels of the whole plant and of the middle leaves illuminated with different light spectra (N = 10 for each light color). Lower case letters indicate *t*-test calculated statistical significance when *p* < 0.05 was assumed.

**Figure 7 plants-14-01334-f007:**
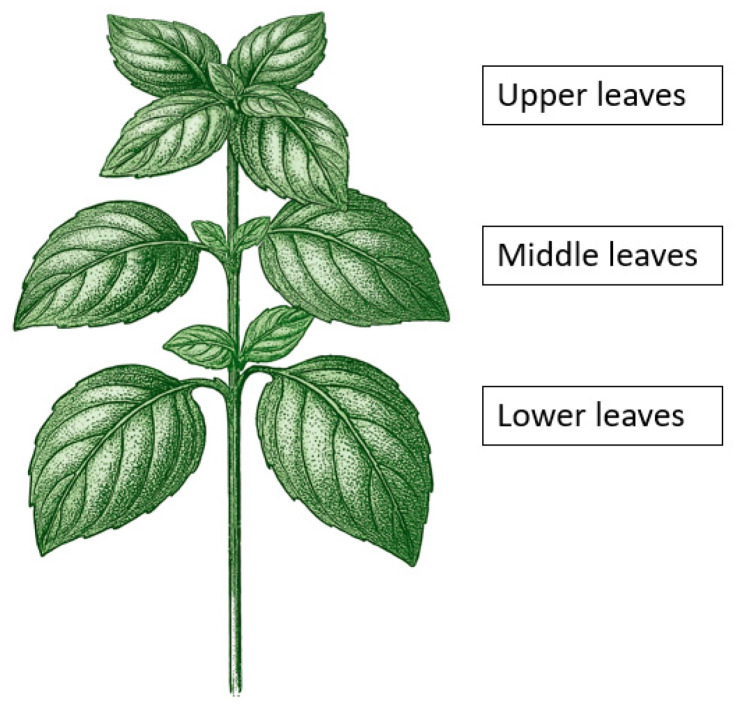
Schematic representation of a typical plant of *Ocimum basilicum* L., cultivar Genoversa, used in the experiments, with marked leaf levels.

**Figure 8 plants-14-01334-f008:**
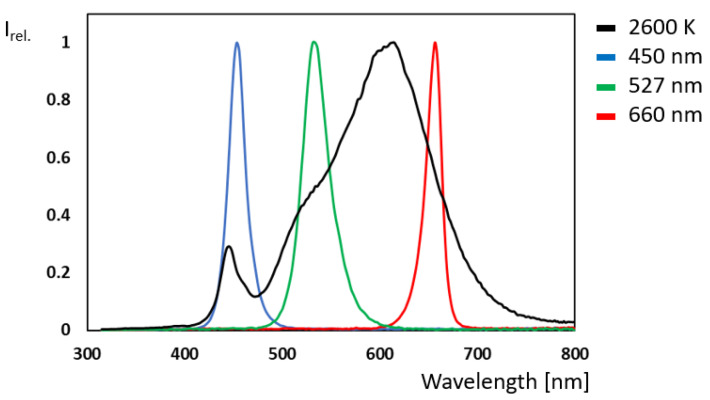
Spectra (relative intensity versus wavelength in nm) of the four seven-engine modules used from Chips 4 Light GmbH. In black, the LA AT020WWG module (correlated color temperature (CCT) = 2600 K); in blue, the LA AT020HBH module (450 nm); in green, the LA AT020SGH module (527 nm); and in red, the LA AT020HRE module (660 nm). All wavelength data denote the peak wavelength, respectively.

## Data Availability

The data that support the findings of this study are openly available at https://github.com/JokicLu/Dataset-.git (accessed on 26 April 2025).

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
