# Peer review of "Effect of Light Intensity and Light Spectrum of LED Light Sources on Photosynthesis and Secondary Metabolite Synthesis in Ocimum basilicum"

_plants, 2025, doi:10.3390/plants14091334_

Round 1

Reviewer 1 Report

Comments and Suggestions for Authors

The authors reported the effects of light-intensity and light-spectrum of LED light sources on photosynthesis and secondary metabolite synthesis in ocimum basilicum. It is valuable and wonderful and should be accepted. I have two questions to communicate.

  1. Did the author completely avoid sunlight when choosing blue, green and red-light sources?
  2. The author chose blue light, green light and red-light source respectively, why did not simultaneously choose the blue light and red light and adjust their proportion.

Author Response

Dear Reviewer,
Thank you very much for taking the time to review our manuscript. We appreciate your thoughtful and constructive feedback. Below we provide a comment and response system, explaining in detail what we have changed. Major changes include the addition of a transitional paragraph to smooth an abrupt shift in the introduction in lines 44-59 and an additional schematic representation of sampling sites in the Materials and Methods section on page 10, lines 356-362. This additional figure changes the number of figures (mostly in the supplementary section), so I've uploaded a new supplementary figures document as well as a new zip file containing each enlarged figure. Finally, we had to update Figure 4 due to a graphical error, which had no effect on the data or its explanation/ interpretation whatsoever.

Comment 1: Did the author completely avoid sunlight when choosing blue, green and red-light sources?

Response 1: Thank you very much for your feedback. We did indeed avoid sunlight during the measurements. Each measurement/ illumination took place in a dark phytochamber with the same greenhouse conditions as for cultivation, but with the expectation that we kept the light intensity at 0 µmol/m2s.

We did add in the Materials and Method section, on page 10, line 374-375, 382 and 389 that the measurements/ illuminations took place in a dark phytochamber.

Comment 2: The author chose blue light, green light and red-light source respectively, why did not simultaneously choose the blue light and red light and adjust their proportion.

Response 2:  The main aim of this study was to understand how the primary light colors of an RGB-spectrum, with the addition of white light, would affect photosynthesis and secondary processes. To continue this research, additional experiments focusing on color mixing are already planned.

Reviewer 2 Report

Comments and Suggestions for Authors

The manuscript has investigated the impact of high light intensities from blue, green, red, and white light on basil’s photosynthetic efficiency, primary and secondary metabolism. It showed that photosynthesis, photoinhibition and secondary metabolite production were wavelength-dependent, broadening our understanding of photosynthesis. Overall, I think these findings will provide valuable information to the field. However, there are some issues that need to be addressed before the manuscript is acceptable.

Major issue:

  1. The introduction section lacks logical coherence, and should be improved for better contextual transitions. For example, in Line 36-44, the shift from ‘basil as a high-value crop’ to ‘basil growth and production of secondary metabolites in response to light has been studied extensively’ is abrupt and lacks continuity. It is recommended to add a transitional paragraph that highlights the importance of light in improving both basil yield and secondary metabolism.

Minor issues:

  1. Line 185, should ‘data not shown’ be Figure 9 ?
  2. Figure 6A, the bottom frame line in Figure 6A is incomplete; Figure 6B, since all comparisons are not significant, the significance label in the figure is unnecessary, but should be clearly mentioned in the figure notes or text.
  3. Line 201-203, it is recommended to add a schematic representation of the sampling sites of the upper, middle, and lower leaves, with additional descriptions in Materials and Methods.
  4. Line 361, it’s better to enclose ‘youngest, directly illuminated leaves’ in parentheses.

Author Response

Dear Reviewer,
Thank you very much for taking the time to review our manuscript. We appreciate your thoughtful and constructive feedback. Below we provide a comment and response system, explaining in detail what we have changed. Major changes include the addition of a transitional paragraph to smooth an abrupt shift in the introduction in lines 44-59 and an additional schematic representation of sampling sites in the Materials and Methods section on page 10, lines 356-362. This additional figure changes the number of figures (mostly in the supplementary section), so I've uploaded a new supplementary figures document as well as a new zip file containing each enlarged figure. Finally, we had to update Figure 4 due to a graphical error, which had no effect on the data or its explanation/ interpretation whatsoever.

Major issue:

Comment 1: The introduction section lacks logical coherence, and should be improved for better contextual transitions. For example, in Line 36-44, the shift from ‘basil as a high-value crop’ to ‘basil growth and production of secondary metabolites in response to light has been studied extensively’ is abrupt and lacks continuity. It is recommended to add a transitional paragraph that highlights the importance of light in improving both basil yield and secondary metabolism.

Response 1: Thank you very much for your feedback and for noticing the abrupt shift in the introduction. We have added the recommended transition paragraph on page 2, lines 44 - 59. We have kept this paragraph short because the main focus of the introduction should be on actual photosynthesis research in basil. We have highlighted what we believe to be a comprehensive and well-written review. It contains more information, if the reader is interested, on how light affects growth/ biomass accumulation and secondary metabolite production.

Minor issues:

Comment 1: Line 185, should ‘data not shown’ be Figure 9 ?

Response 1: We agree that "data not shown" is actually Figure 9 (now 10). As Figures 6 and 9 look almost identical, and in order not to confuse the reader or interrupt the reading flow, we have decided to include Figure 9 only in the "Supplementary Figures".
On page 6, Line 207 we updated “data not shown” in “Figure 10 A, B; Supplementary figures”

Comment 2: Figure 6A, the bottom frame line in Figure 6A is incomplete; Figure 6B, since all comparisons are not significant, the significance label in the figure is unnecessary, but should be clearly mentioned in the figure notes or text.

Response 2:  We are sorry but we can’t identify where the bottom line of Figure 6A is incomplete. Thank you for pointing out that the significance label in Figure 6 B is unnecessary. However, in order to ensure a uniform figure scheme and reading flow, we would like to keep the present figure.

Comment 3: Line 201-203, it is recommended to add a schematic representation of the sampling sites of the upper, middle, and lower leaves, with additional descriptions in Materials and Methods.

Response 3: We added a schematic representation of the sampling sites and added an extra description on page 10 line 356-362

Comment 4: Line 361, it’s better to enclose ‘youngest, directly illuminated leaves’ in parentheses.

Response 4: We followed the recommendation and updated the paratheses, (now line 391).

Reviewer 3 Report

Comments and Suggestions for Authors

The manuscript is well written and presents interesting results on the influence of some light parameters on photosynthesis in basil plants. It can be improved in the presentation of some aspects, as highlighted in the text. 

  1. Please organise the spelling of the units according to the SI system.
    2. Please note that there is a space between the unit and the number.
    3. Latin names of plants are written in italics.
    4. In the figures of the parameters of photosynthesis, the authors pay attention only to the plateau point. However, the plateau is often followed by a period of stagnation in which the value changes very little. It would also be important to pay attention to this, because often the maximum or minimum for the parameter being studied starts earlier. Or the values that occur earlier or after the plateau are not significantly different from this point.More comments in the article. 

Author Response

Dear Reviewer,
Thank you very much for taking the time to review our manuscript. We appreciate your thoughtful and constructive feedback. Below we provide a comment and response system, explaining in detail what we have changed. Major changes include the addition of a transitional paragraph to smooth an abrupt shift in the introduction in lines 44-59 and an additional schematic representation of sampling sites in the Materials and Methods section on page 10, lines 356-362. This additional figure changes the number of figures (mostly in the supplementary section), so I've uploaded a new supplementary figures document as well as a new zip file containing each enlarged figure. Finally, we had to update Figure 4 due to a graphical error, which had no effect on the data or its explanation/ interpretation whatsoever.

Comment 1: Please organise the spelling of the units according to the SI system.
Response 1: Thank you very much for your detailed feedback. We updated the spelling of units according to the SI-system.

Comment 2: Please note that there is a space between the unit and the number.
Response 2:
  We also added space in-between number and units.

Comment 3: Latin names of plants are written in italics.
Response 3:
Latin names were also updated to italics

Comment 4: In the figures of the parameters of photosynthesis, the authors pay attention only to the plateau point. However, the plateau is often followed by a period of stagnation in which the value changes very little. It would also be important to pay attention to this, because often the maximum or minimum for the parameter being studied starts earlier. Or the values that occur earlier or after the plateau are not significantly different from this point.

Response 4:
In our opinion, these key points of a saturating assimilation curve are discussed. Maximum assimilation is mentioned, as well as when the plateau begins and ends. We further discuss that the end of the plateau means that there are likely photoinhibitory effects that lead to a decrease in assimilation.

Comment 5: (from the attached reviewed script) Please don't use the same words as in the title

Response 5: Thank you for the advice to reach a wider audience by not matching the keywords to the title. We have changed two keywords accordingly.

Comment 6: (from the attached reviewed script) Please enlarge the figures to the full length of the page

Response 6:
We apologize for the oversight regarding the figure editing. We were not aware that it was our responsibility to make these adjustments, as we had assumed that the journal would handle this part of the production process

Comment 7: (from the attached reviewed script) This paragraph only creates confusion, it can be partly moved to methods and discussion.

Response 7: We believe that this paragraph provides a good introduction to why and how these experiments were conducted and how they relate to the others. When written in materials and methods, we are afraid it is not conclusive as to why we did these experiments.

Comment 8: (from the attached reviewed script) This section, like the one above, creates confusion. The data for 2,500 micromoles are included in the supplement and so please write, not that they are unpublished. Please remember that this is a results section and not a discussion.

Response 8: Thank you for noticing. We have paraphrased the first sentence to clarify that we did a different experiment, so there is no confusion or mixing of experiments.
We also agree that "data not shown" is misleading and is actually Figure 9 (now 10). As Figures 6 and 9 look almost identical, and in order not to confuse the reader or interrupt the reading flow, we have decided to include Figure 9 only in the "Supplementary Figures".
On page 6, Line 207 we updated “data not shown” in “Figure 10 A, B; Supplementary figures”
We have also cut the last sentence of this paragraph, as it already discusses the results

Comment 9: (from the attached reviewed script) Conclusion:
In this sentence, the authors suggest that post-harvest treatment of plants can increase phenolic and flavanoid content. Please write this more precisely, what the authors meant when they wrote about treating plants with blue light during this period.

Response 9:
 The conclusion was updated; lines 333-338.

Comment 10: (from the attached reviewed script) Conclusion: Please write more precisely and refer to each color

Response 10:  In the conclusion, our aim was to summaries each result and discussion in the shortest possible way. To list them all again would be to miss the point. The results are listed in detail in the results section.

Comment 11: (from the attached reviewed script) How were the authors able to maintain a 12-hour day under greenhouse conditions throughout the growing period? When were the plants grown in the greenhouse?

Response 11: Although our plants grew in a greenhouse, the effect of sunlight was not significant. The greenhouse is located between a forest and a university building, so the primary light source is our greenhouse LED light source. These are controlled so that they have a 12-hour day and night regulation.